# Modelling to explore the potential impact of asymptomatic human infections on transmission and dynamics of African sleeping sickness

**Maryam Aliee**[1,2]*, **Matt J. Keeling**[1,2,3], **Kat S. Rock**[1,2]

**1** Mathematics Institute, University of Warwick, Coventry, United Kingdom, **2** Zeeman Institute for Systems Biology and Infectious Disease Epidemiology Research, University of Warwick, Coventry, United Kingdom, **3** School of Life Sciences, University of Warwick, Coventry, United Kingdom

* maryam.aliee@warwick.ac.uk

**Data Availability Statement:** All relevant data are within the manuscript and its Supporting information files.

## Abstract

*Gambiense* human African trypanosomiasis (gHAT, sleeping sickness) is one of several neglected tropical diseases (NTDs) where there is evidence of asymptomatic human infection but there is uncertainty of the role it plays in transmission and maintenance. To explore possible consequences of asymptomatic infections, particularly in the context of elimination of transmission—a goal set to be achieved by 2030—we propose a novel dynamic transmission model to account for the asymptomatic population. This extends an established framework, basing infection progression on a number of experimental and observation gHAT studies. Asymptomatic gHAT infections include those in people with blood-dwelling trypanosomes, but no discernible symptoms, or those with parasites only detectable in skin. Given current protocols, asymptomatic infection with blood parasites may be diagnosed and treated, based on observable parasitaemia, in contrast to many other diseases for which treatment (and/or diagnosis) may be based on symptomatic infection. We construct a model in which exposed people can either progress to either asymptomatic skin-only parasite infection, which would not be diagnosed through active screening algorithms, or blood-parasite infection, which is likely to be diagnosed if tested. We add extra parameters to the baseline model including different self-cure, recovery, transmission and detection rates for skin-only or blood infections. Performing sensitivity analysis suggests all the new parameters introduced in the asymptomatic model can impact the infection dynamics substantially. Among them, the proportion of exposures resulting in initial skin or blood infection appears the most influential parameter. For some plausible parameterisations, an initial fall in infection prevalence due to interventions could subsequently stagnate even under continued screening due to the formation of a new, lower endemic equilibrium. Excluding this scenario, our results still highlight the possibility for asymptomatic infection to slow down progress towards elimination of transmission. Location-specific model fitting will be needed to determine if and where this could pose a threat.

**Funding:** This work was supported by the Bill and Melinda Gates Foundation (www.gatesfoundation. org) through the NTD Modelling Consortium [OPP1184344] (M.A., M.J.K. and K.S.R.) and the Human African Trypanosomiasis Modelling and Economic Predictions for Policy (HAT MEPP) project [OPP1177824] (M.J.K. and K.S.R.). The funders had no role in study design, data collection and analysis, decision to publish, or preparation of the manuscript.

**Competing interests:** The authors have declared that no competing interests exist.

## Author summary

*Gambiense* African sleeping sickness is an infectious disease targeted for elimination of transmission by 2030. Despite this there is still some uncertainty how frequently some infected people who may not have symptoms could "self-cure" without ever having disease and whether some types of infections, such as infections only in the skin, but not the blood, could still contribute to transmission, yet go undiagnosed. To explore how problematic these asymptomatic infections could be in terms of the elimination goal, we use a mathematical model which quantitatively describes changes to infection and transmission over time and includes these different types of infection. We use results of published experimental or field studies as inputs for the model parameters governing asymptomatic infections. We examined the impact of asymptomatic infections when control interventions are put in place. Compared to a baseline model with no asymptomatics, including asymptomatic infection using plausible biological parameters can have a profound impact on transmission and slow progress towards elimination. In some instances it could be possible that even after initial decline in sleeping sickness cases, progress could stagnate without reaching the elimination goal at all, however location-specific modelling will be needed to determine if and where this could pose a threat.

## Introduction

*Gambiense* human African trypanosomiasis (gHAT, sleeping sickness) is one of the often fatal neglected tropical diseases [1, 2], with the causative parasite, *Trypanosoma brucei gambiense*, transmitted to humans through the a bite of tsetse (*Glossina*). Transmission of gHAT has reduced significantly in the recent years—from 37,385 globally reported cases in 1998 to less that 1000 in 2019—thanks to a range of interventions in endemic areas [3].

Tsetse are obligate blood-feeders and biological transmission of the parasite to humans is usually considered to be through the injection trypanosomes into blood during feeding. Trypanosomes can replicate in the blood and this is why gHAT infection is associated with presence of parasite in blood [4]. However, various studies, in experimental animal models and recently in humans, observed localised aggregation of trypanosomes in the skin matrix [5–8]. Further animal experiments observe accumulation of sequestered trypanosomes in other organs such as the spleen, liver, brain, and extravascular adipose tissue [9–11]. The existence of parasites in skin has even been reported when there is an absence of detectable parasites in blood. These "skin-only" infections are postulated to cause infections which may spontaneously disappear without any treatment, or alternatively develop into a blood infection [12, 13]. Whilst most infection to vectors may be acquired through human blood, skin and perhaps adipose tissues are believed to play a role in transmission as well [5–7, 11].

Even putting aside skin-only infections, there may be other types of asymptomatic human infection that could alter transmission dynamics. The natural history of gHAT infection is relatively long in duration, with early infections often having minimal symptoms. A longitudinal study in Côte d'Ivoire has previously found that some people with confirmed infections (with detectable blood trypanosomes) that refused treatment, subsequently tested negative for infection indicating that some may have self-cured rather progress to the more severe, late-stage disease [14].

The current diagnostic pathway includes people first being serologically screened using finger-prick blood taken either during mass (active) screening in at-risk village, or in passive health facilities if the patient is considered a suspect based on gHAT-like symptoms. Positive serological suspects needs to be confirmed by microscope observation of trypanosomes in blood, lymph or cerebrospinal fluid (CSF) [15]. This protocol does allow for identification and treatment of people asymptomatically infected with blood parasites. Conversely there are some blood infections that might be missed—due to good but imperfect diagnostic algorithm sensitivity—and, furthermore, skin-only infections cannot be diagnosed within the current protocols. Previous studies reported some seropositive individuals remain without a confirmed parasitological diagnosis for years [16]. Similarly, there is almost no chance to find asymptomatic (or mildly symptomatic) infections (of any type) in passive screening, which relies largely on symptoms to motivate the patient to seek care and as an enter point to be screened by health care staff [17].

Despite imperfect tools, and the challenges outlined above, the global drive to control gHAT has resulted in huge declines in reported cases and gHAT is therefore targeted by the World Health Organization (WHO) for elimination of transmission (EOT) by 2030 [18]. The elimination of gHAT would be a tremendous achievement, as it would for any infectious disease, however there are still a number of hurdles to overcome in the road to elimination. Two such hurdles are detailed in Büscher *et al* [19]: specifically (i) the potential of either non-human reservoir animals to play a role in the transmission and/or maintenance of infection to humans and (ii) the existence of asymptomatic human infections which may or may not be detectable in routine screening for infection and could have the possibility to self-cure. Although the role of animal transmission is far from resolved, there are a number of quantitative modelling studies which have previously attempted to understand the possible impact that animals could have on achievement of the elimination goal [20–25]. Yet only two articles [12, 26] present explicit models of asymptomatic human infection with the possibility of self-curing infections. Whilst one [12] quantifies the likely contribution to transmission of asymptomatics in the Forécariah focus of Guinea, and the other is used for predicting timelines until elimination of transmission in former Bandundu province of DRC, there are still several outstanding questions about how asymptomatic human infections could impact the trajectory to elimination under different intervention strategies. Therefore the importance of asymptomatic infections for the spread of the infection remains a relevant concern. The current incidence of case reporting across Africa have now fallen to historic lows, leading to some optimism that elimination is within sight. Despite this, asymptomatic infection as a driver of transmission could be worrisome, particularly considering the previous resurgence of the gHAT that happened between 1970–1990 after it dipped to low levels [27].

In this manuscript, we study the potential impact of asymptomatic infection which may self-cure on gHAT dynamics using a mathematical model. In this novel model extension we include the possibility that skin-only parasite infections contribute to transmission. Whilst other anatomical reservoirs have been identified, we do not include them here. Our model is based on a previously established compartmental model developed to study the transmission and medical interventions [22, 25, 26, 28–31]. In the next section we present a minimal model to explicitly account for human skin-only and blood infections in this compartmental gHAT model. We analyse how the endemic equilibrium and basic reproduction number change in this extended model by applying sensitivity analysis and exploring parameter space. Subsequently we examine the impact on transmission dynamics over time as medical interventions are applied to analyse the theoretical ramifications of asymptomatic human infection on achievement of the 2030 goal.

## Materials and methods

The novel model presented here to study asymptomatic infections of gHAT dynamics is based on a baseline model presented in Rock *et al.* [22] and with various adaptations made, resulting in the latest version presented in Crump *et al.* [30]. The baseline model takes into account different subpopulations of humans and tsetse, represented by compartments of different infection states. Humans can be exposed and subsequently become infectious through a bite of an infectious tsetse. They progress through stage 1 (early stage) and stage 2 (late stage) of disease with specific rates ($\eta_H(Y)$ and $\gamma_H(Y)$—dependent on the year, $Y$—respectively). Tsetse vectors can become exposed and subsequently infectious if they bite an infectious human. Infected people may be detected by passive or active screening, followed by treatment and recovery. Here, we consider a version of the baseline model where humans are partitioned into two subgroups of (i) low-risk and participating in the active screening, and (ii) high-risk and non-participating in active screening (full details are presented in Crump *et al* [30]).

To account for asymptomatic infections, we modify this model by considering two subgroups within the first stage infected humans, labeled as skin parasite $I_{1H}^s$ and blood parasite $I_{1H}^b$ populations (Fig 1). Therefore, exposed population are assumed to develop either parasite infection with detectable levels of parasites in the blood (with probability $p_{bs}$) or skin-only parasite infection (with probability $1 - p_{bs}$). We assume skin-only infections are asymptomatic, and would not be diagnosed in active screening within the current protocols due to the lack of parasite in their blood (even if these infected people may test positive in initial screening tests such as the card agglutination test for trypanosomes (CATT) or in rapid diagnostic tests (RDTs) based on antibody expression). We take into account asymptomatic cases in blood parasite group as well. As the main difference, this group will be likely diagnosed in active screening if tested (according to the sensitivity of the algorithm used—which is usually considered to be over 90%). In the novel model, we add the possibility of self-cure for both the skin-only and blood parasite groups through the parameters $\omega_H^b$ for blood and $\omega_H^s$ for skin-only. Putting everything together, we modify the dynamic equations of the baseline model for humans to yield:

$$
\begin{aligned}
\frac{dS_{Hi}}{dt} &= \mu_H N_{Hi} - \mu_H S_{Hi} + \omega_H R_{Hi} + \omega_H^s I_{1Hi}^s + \omega_H^b I_{1Hi}^b - \alpha m_{\text{eff}} f(i) \frac{S_{Hi}}{N_{Hi}} I_V \\
\frac{dE_{Hi}}{dt} &= \alpha m_{\text{eff}} f(i) \frac{S_{Hi}}{N_{Hi}} I_V - (\sigma_H + \mu_H) E_{Hi} \\
\frac{dI_{1Hi}^s}{dt} &= (1 - p_{bs}) \sigma_H E_{Hi} - (\omega_H^s + \theta + \mu_H) I_{1Hi}^s \\
\frac{dI_{1Hi}^b}{dt} &= p_{bs} \sigma_H E_{Hi} - (\varphi_H + \mu_H + \eta_H(Y) + \omega_H^b) I_{1Hi}^b + \theta I_{1Hi}^s \\
\frac{dI_{2Hi}}{dt} &= \varphi_H I_{1Hi}^b - (\gamma_H(Y) + \mu_H) I_{2Hi} \\
\frac{dR_{Hi}}{dt} &= \eta_H(Y) I_{1Hi}^b + \gamma_H(Y) I_{2Hi} - (\omega_H + \mu_H) R_{Hi}.
\end{aligned}
\tag{1}
$$

The subscript *i* denotes that human subpopulations of either high-risk or low-risk. The dynamics of these continuous ODEs are identical for both risk groups except for the probability of being bitten by tsetse, $f(i)$ which is *r*-fold higher for the high-risk group, and when we simulate the non-continuous active screening interventions, which is assumed to take place at the beginning of each year, during which some infected individuals may be identified and treated. Following analysis of the model dynamics in DRC and Chad, we try to capture the

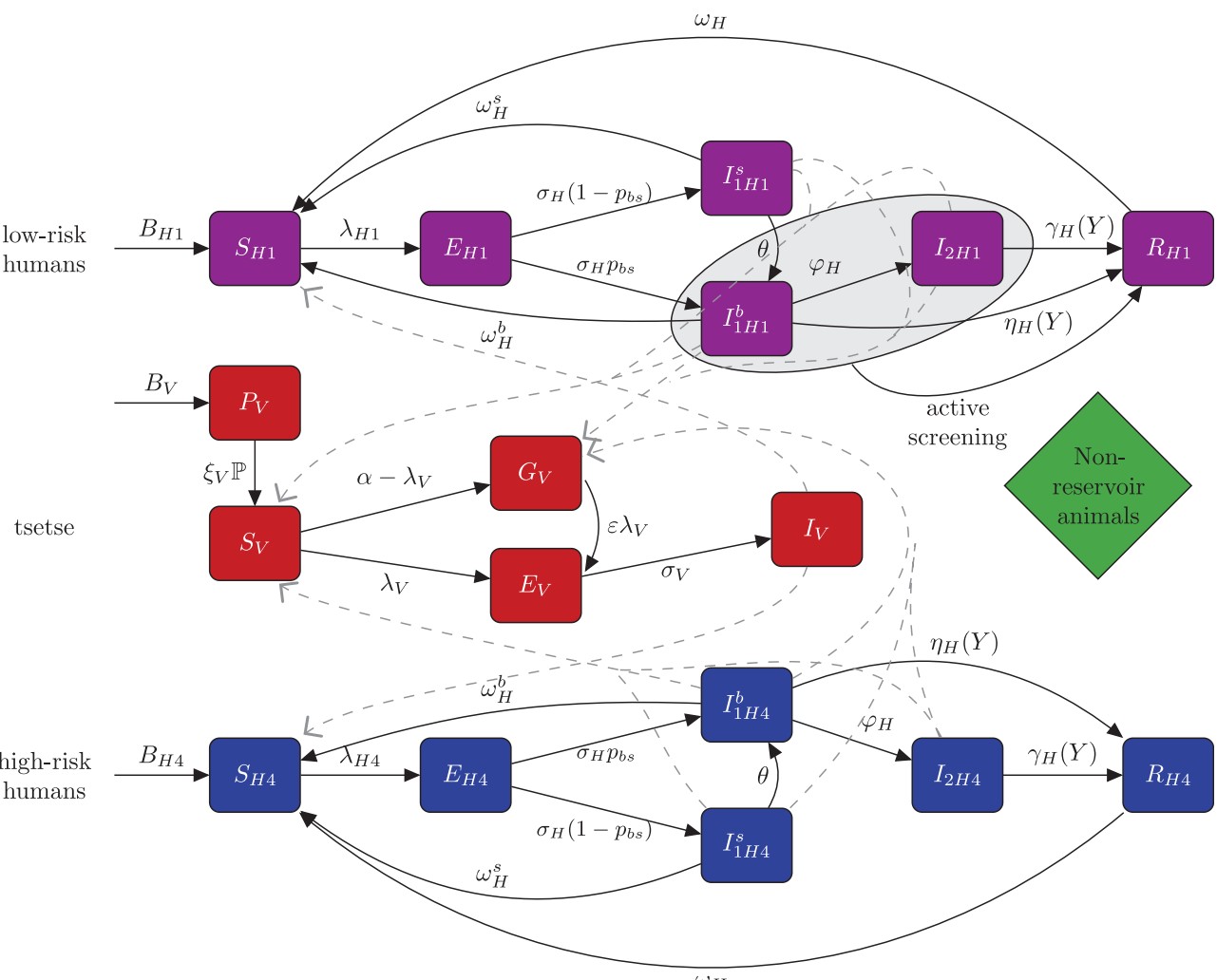

**Fig 1. Schematic of the model to describe gHAT transmission with asymptomatic infections.** This multi-host model of HAT takes into account high- and low-risk groups of humans and their interactions with tsetse vectors. Each group consists of different compartments: Susceptible humans $S_{Hi}$ can become exposed on a bite of an infectious tsetse. Exposed people $E_{Hi}$ progress to become the skin-only parasite $I_{1Hi}^s$ or blood parasite stage 1 infection $I_{1Hi}^b$; the latter eventually develops stage 2 (if not detected in screening), and once treated they recover by hospitalisation $R_{Hi}$. Active screening can accelerate treatment rate of infected people, but only in those with detectable blood infection and in the low-risk group—$I_{1H1}^b$ and $I_{2H1}$. This is marked on the diagram as a grey oval. Here we assume high-risk group does not participate in active screening. By biting an infectious person, tsetse can become exposed and subsequently infectious, $E_V$ and $I_V$. $G_V$ represents the tsetse population not exposed to *Trypanosoma brucei gambiense* in the first blood-meal and are therefore less susceptible in the following meals. Rates are shown by Greek letters associated with arrows. An animal reservoir is not considered. This figure is a modified version of the original one [22, 30].

effect of certain subpopulations (e.g. working age males) having lower participation in screening but also being most at risk from exposure to tsetse bites, and therefore assume only low risk individuals may present in active screening which occurs randomly and according to the coverage in each year [22, 25].

This model accounts for different compartments within each group to describe infection dynamics (Fig 1). Susceptible humans $S_{Hi}$ can become exposed on a bite of an infectious tsetse. Exposed people $E_{Hi}$ develop either the skin-only parasite $I_{1Hi}^s$ or blood parasite stage 1 infection $I_{1Hi}^s$; the latter eventually develops stage 2 if not detected in screening. We assume skin infections can transform to blood infections with a rate indicated by θ. Recovered populations $R_{Hi}$

include all people treated in screening programs. The rates of transitions between compartments are described by parameters listed in Table 1.

Passive screening is considered in the equations with the rates proportional to $\eta_H(Y)$ and $\gamma_H(Y)$ corresponding to the first and second stages of the disease. Before 1998 (pre-active screening) it was assumed that passive detection was less effective due to lack of good diagnostic test availability, and only so identified stage 2 individuals at a rate $u(1997)\gamma_H^{\text{pre}}$, which is smaller than the stage 2 passive detection rate in 1998, $u(1998)\gamma_H^{\text{post}}$. Following previous modelling work using gHAT data from former Bandundu province, there is a strong signal from epidemiological staging data that passive screening has improved during the time period from 2000–2012 [26, 30]. To capture the improvement of stage 1 to stage 2 passive detection, the model utilises the following formula:

$$\eta_H(Y) = \eta_H^{\text{post}}\left[1 + \frac{\eta_{H\text{amp}}}{1 + \exp(-d_{\text{steep}}(Y - d_{\text{change}}))}\right],$$

$$\gamma_H(Y) = \gamma_H^{\text{post}}\left[1 + \frac{\gamma_{H\text{amp}}}{1 + \exp(-d_{\text{steep}}(Y - d_{\text{change}}))}\right],$$

where $Y$ is the year and $\eta_H(Y)$ is the annual stage 1 passive detection rate. Parameters dictating the amplitude, steepness and switching year can be found in Table 1. $u(Y)$ is calculated accordingly to keep the death rate $(1 - u(Y))\gamma_H(Y)$ fixed.

Equations for vector dynamics are very similar to the baseline model, with the only change occurring in the ability to take a meal from the different subpopulations of infected people:

$$\frac{dP_V}{dt} = B_V N_H - (\xi_V + \frac{P_V}{K})P_V$$

$$\frac{dS_V}{dt} = \xi_V \mathbb{P}(\text{pupating})P_V - \alpha S_V - \mu_V S_V$$

$$\frac{dE_{1V}}{dt} = \alpha p_V \sum_i f(i) \frac{(I_{1Hi}^b + x I_{1Hi}^s + I_{2Hi})}{N_{Hi}}(S_V + \varepsilon G_V) - (3\sigma_V + \mu_V)E_{1V}$$

$$\frac{dE_{2V}}{dt} = 3\sigma_V E_{1V} - (3\sigma_V + \mu_V)E_{2V}$$

$$\frac{dE_{3V}}{dt} = 3\sigma_V E_{2V} - (3\sigma_V + \mu_V)E_{3V} \tag{2}$$

$$\frac{dI_V}{dt} = 3\sigma_V E_{3V} - \mu_V I_V$$

$$\frac{dG_V}{dt} = \alpha(1 - p_V \sum_i f(i) \frac{(I_{1Hi}^b + x I_{1Hi}^s + I_{2Hi})}{N_{Hi}})S_V$$

$$- \alpha p_V \varepsilon \sum_i f(i) \frac{(I_{1Hi}^b + x I_{1Hi}^s + I_{2Hi})}{N_{Hi}}G_V - \mu_V G_V$$

As shown in Fig 1 and similar to the baseline model, this model considers different compartments of tsetse, describing the pupal stage $P_V$ that can develop adults, which form susceptible vectors $S_V$. By biting an infectious person, tsetse can become exposed and subsequently infectious, $E_V$ and $I_V$. $G_V$ represents the tsetse population not exposed to *Trypanosoma brucei gambiense* in the first blood-meal with reduced susceptibility in the following meals. The actual number the total population of adult tsetse is normalised to match the number of humans $N_H$, through a non-dimensionalisation process outlined in Rock *et al.* [22] to reduce the number of

**Table 1. Model parameters.** Notation, description, and a range of all parameters used for sensitivity analysis. We split the list by those parameters that are typically considered as fixed in previous analysis of the baseline model, those that are featured in the baseline model but are fitted to data, and the five new parameters that are new to this asymptomatic model variant. We also show the one with the highest likelihood for the health zone Mosango.

| Notation | Description | Range | Unit |
|---|---|---|---|
| **Fixed parameters** | | | |
| $\mu_H$ | Natural human mortality rate | 5.48 [4.3,6.1] $\times$ $10^{-5}$ | days$^{-1}$ |
| $\sigma_H$ | Human incubation rate | 0.0833 [0.03, 0.5] | days$^{-1}$ |
| $\varphi_H$ | Stage 1 to 2 progression rate | 0.0019 [0.001, 0.003] | days$^{-1}$ |
| $\omega_H$ | Recovery rate/waning-immunity rate | 0.006 [0.002, 0.01] | days$^{-1}$ |
| Sens | Active screening diagnostic sensitivity | 0.91 [0.85, 0.99] | - |
| $B_V$ | Tsetse birth rate* | - | days$^{-1}$ |
| $\xi_V$ | Pupal death rate | 0.037 [0.02, 0.05] | days$^{-1}$ |
| $K$ | Pupal carrying capacity | =111.09 [80, 200]$N_H$ | |
| $\mathbb{P}$ | Probability of pupating | 0.75 [0.6, 0.9] | |
| $\mu_V$ | Tsetse mortality rate | 0.03 [0.014, 0.047] | days$^{-1}$ |
| $\sigma_V$ | Tsetse incubation rate | 0.034 [0.025, 0.1] | days$^{-1}$ |
| $\alpha$ | Tsetse bite rate | 0.333 [0.2, 0.5] | days$^{-1}$ |
| $p_V$ | Probability of tsetse infection per single infective bite | 0.065 [0.05, 0.08] | - |
| $\varepsilon$ | Reduced non-teneral susceptibility factor | 0.05 [0.02, 0.08] | - |
| $f_H$ | Proportion of blood-meals on humans | 0.09[0.05, 1] | - |
| **Fitted parameters** | | | |
| $R_0$ | Basic reproduction number | 1.015 [1.001, 1.3] | - |
| $m_{eff}$ | Effective tsetse density which is equal to the probability of human infection from a single blood meal by an infectious tsetse times the tsetse to human population density*. | - | - |
| $r$ | Relative bites taken on high-risk humans | 3.322 [1.1, 20] | - |
| $k_1$ | Proportion of low-risk people | 0.909 [0.5, 0.99] | - |
| Spec | Active screening diagnostic specificity | 0.9994 [0.996, 1] | - |
| $u(1998)$ | Proportion of passive cases reported in 1998 | 0.276 [0.2, 0.8] | - |
| $\gamma_H^{\mathrm{pre}}$ | Treatment/death rate from stage 2 (pre-1998) | 1.91 [0.1, 6] $\times$ $10^{-3}$ | - |
| $\gamma_H^{\mathrm{post}}$ | Treatment/death rate from stage 2 (post-1998) | 2.57 [1, 2] $\times$ $10^{-3}$ | - |
| $\eta_H^{\mathrm{post}}$ | Treatment rate from stage 1 (post-1998) | 0.864 [0.1, 3] $\times$ $10^{-4}$ | days$^{-1}$ |
| $\eta_{H_{\mathrm{amp}}}$ | Relative improvement in passive stage 1 detection rate | 0.752 [0.1, 5] | - |
| $\gamma_{H_{\mathrm{amp}}}$ | Relative improvement in treatment/death rate from stage 2 | 0.287 [0.01, 1] | - |
| $d_{\mathrm{steep}}$ | Speed of improvement in passive detection | 1.19 [0.5, 1.8] | - |
| $d_{\mathrm{change}}$ | Midpoint year for passive improvement | 2004.3 [2002, 2012] | - |
| **Asymptomatic parameters** | | | |
| $p_{bs}$ | Proportion of exposures resulting in initial blood infection | 0.8 [0.4, 0.99] | - |
| $\omega_H^b$ | Self-cure rate of blood infections | 1 [0, 10] $\times$ $10^{-5}$ | days$^{-1}$ |

*(Continued)*

**Table 1.** (Continued)

| Notation | Description | Range | Unit |
|---|---|---|---|
| $\omega_H^s$ | Self-cure rate of skin-only infections | $6\,[1, 10] \times 10^{-4}$ | $days^{-1}$ |
| $\theta$ | Transition rate from skin-only infection to blood infection | $2.5\,[1, 5] \times 10^{-4}$ | $days^{-1}$ |
| $x$ | Relative infectiousness of a skin-only infection compared to a blood infection | $0.5\,[0, 1]$ | - |

*This parameter is not directly fitted but post-calculated after fitting $R_0$

free parameters by one. We assumed the skin infection is less transmissible compared to the blood infection, implemented with the coefficient $x$.

A full list of model parameter descriptions is found in Table 1. Included in this list is the bundled parameter, $R_0$, which is the basic reproduction number of the baseline model, defined by the next generation matrix approach [32] and describes average numbers of secondary infections from an average infected individual (human or tsetse) in an otherwise susceptible population. This parameter is important as it has regularly been used to aid model fitting to data [22, 30]. A full derivation of $R_0$ in the asymptomatic model is given in "S1 Methods".

## Estimating model parameters

This model is characterised by the parameters listed in Table 1. It is important to use the appropriate values of the parameters, however, many of them are difficult to estimate or could vary by location. The baseline model contains two sets of parameters, fixed and fitted ones. Fixed parameters are estimated based on available data in the literature, however, there is still uncertainty about their exact values. For instance, natural human mortality rate, $\mu_H$, is estimated based on World Bank reports on life expectancy, which changes between different gHAT-endemic regions and also over time [33]. We therefore come up with an accepted range for each fixed parameter shown Table 1.

Fitted parameters default values are taken from posterior distributions by fitting the model to the health-zone-level data for different health zones of the Democratic Republic of Congo (DRC) using an adaptive Metropolis-Hastings Markov chain Monte Carole (MCMC) algorithm. In previous studies, informative priors were used for fitting parameters based on some estimates available in the literature or values coming from fitting the model to data at the province level [22, 30]. In the present study we use these priors to inform the possible range that parameters might take and expand some of them to account for possible fluctuation since the model structure has now been changed. For instance, the basic reproduction number, $R_0$, could potentially reach higher values even for the same observed case incidence due to the asymptomatic infection compared to the baseline model.

Apart from the parameters of the baseline model, five extra parameters are introduced in our minimal asymptomatic model as discussed in the previous section, indicated by an orange colour in Table 1. Considering the nature of asymptomatic infection, it is not straightforward to quantify the corresponding parameters ranges. Our main purpose in this research is to study sensitivity of the model extension with respect to the asymptomatic parameters, and to avoid overestimating it, we make some assumptions about the biology to constraint the ranges.

For instance, the blood infections are assumed to be the dominant form of infection. Our default value for the proportion was influenced by two studies, one showing skin biopsies from DRC revealed 6/1121 samples were skin parasite positive from people who had no gHAT diagnosis (either clinical signs or blood parasites), during a period when reported gHAT incidence in the area at the time was 1.5–2% [7]. Another study in Guinea shows 40/5418 were CATT

positive among which 28 were confirmed via blood parasitology, while for all 26 CATT positives enrolled in the study (8 unconfirmed and 16 confirmed) trypanosomes were identified in their skin samples via staining by a *T. brucei*-specific antibody [13]. It is important to note that the relative proportions of skin-only and blood infections observed in cross-sectional survey do not alone give the full picture as average time spend infected in each state influences these observations. Considering uncertainties around this parameter including small sample sizes from published studies, we use a fairly broad range for proportion of blood infection $p_{bs}$ to include lower values starting from 40%. Moreover, the transmissibility of skin-only parasite infection is probably lower compared to blood infection (a mouse study found that skin infections without detectable blood parasites were around 60% as likely to infect tsetse compared to skin and blood infections [7]), and we therefore consider relative infectiousness of a skin infection $x$ between 0 and 1.

The other parameters, self-cure rates from skin and blood infection populations as well as transition rate from skin to blood infection population, can have substantial influence on the infection dynamics. In this study we choose these parameters more cautiously based on some estimates in literature. For self cure from blood infections we utilised data from a small sample size study in Côte d'Ivoire—the study followed up patients refusing treatment where, out of 15 people who were originally (blood) parasite positive, nine were parasite negative after around 3.5 years [34]. In [14], out of 53 followed-up patients who had confirmed blood infections and refused treatment in 1995/96, by 1999, nine had become parasite negative, and by 2002 three more were parasite negative. Therefore we might assume that at least 12/53 blood infections result in no symptoms developing. During follow up at least 31 infected people developed symptoms or died due to HAT disease indicating that more people with blood parasites progress to disease rather than self-curing ($\omega_H^b < \eta_H(Y) + \varphi_H + \mu_H$). The self-cure rate from skin-only infections, $\omega_H^s$, can be in part guided by a study in Guinea that followed up 5 unconfirmed CATT positive for whom CATT and skin staining was initially positive, but 4 of these people became negative in CATT and skin biopsy after 6 months [13]. Unfortunately without knowing when initial infection occurred and with a very small number of infections of this type identified it is hard to pinpoint a robust estimate for the self-cure rate, and so here we assume an upper limit of $\omega_H^s$ to be $10^{-3}$. If $\omega_H^s$ was, in fact, even higher, the average time spend infected as a skin-only infection would be less, and therefore there would be lower contribution from these asymptomatics. The final unconfirmed CATT positive individual was still positive after 6 months, and was eventually diagnosed as a stage 1 case during a medical survey almost 2.5 years after enrollment we conclude transition rate from skin-only infection to blood infection, $\theta$, is likely smaller than its self-cure rate $\omega_H^s$.

## Model simulations

The model presented in the previous section can be solved to estimate the potential roles of the asymptomatic infection on the transmission and dynamics of the sleeping sickness. In "S1 Methods", we present analytical equations which give the endemic equilibrium of this ODE model for gHAT. To study dynamics of gHAT over time in our model, we solve the Eqs 1 and 2 numerically with the help of 4th order Runge-Kutta methods using the time-step of 1 day, which is sufficient to achieve high accuracy due to the slow infection dynamics. In these simulations we start from our analytical computed endemic equilibrium, which has the advantage of saving simulation time compared to numerical computation of the steady state. We assume that active screening begins in 1998, and that this is the first year the system is perturbed from steady state. We simulate continuation of active screening at a constant level—30% coverage per year with the perfect specificity—which can only detect the blood parasite infection in low-

risk population (as explained in the beginning of this section). The code used to perform model simulations is included in "S1 Analysis Code".

## Results

Both the endemic equilibrium configuration and dynamic behaviour of the model are characterised by the parameters shown in Table 1. To explore the role of the parameters, we first perform sensitivity analysis of the model. This will be followed by analysing the impact of the additional parameters of the asymptomatic model on the endemic equilibrium and dynamics of the system.

### Sensitivity analysis of the full model

As introduced in Table 1, our compartmental model of gHAT dynamics is characterised by 32 parameters, within which five correspond to asymptomatic infection. As we explained in the methods, there are estimates available for some of the parameters, however, all have some level of uncertainty in their estimates to varying degrees. We then presented a plausible range of each parameter based on available evidence in data, laboratory experiments, or accepted understanding of the circumstances. In this section, we quantify the sensitivity of our model in terms of all the parameters. We particularly analyse the sensitivity of the total prevalence given by

$$I_H/N_H = \frac{1}{N_H}\sum_i I^s_{1Hi} + I^b_{1Hi} + I_{2Hi}$$

at the endemic equilibrium configuration before 1998, $I^0_H/N_H$, and its relative change in 2020, $I_H(2020)/I^0_H$.

To perform the sensitivity analysis, we generate 100,000 random samples of sets of parameters drawn uniformly from the intervals identified in Table 1. We then calculate the prevalence values for each parameter set and calculate the sensitivity values with the help of MATLAB using a ranked partial correlation method (https://uk.mathworks.com/help/sldo/ref/sdo.analyze.html). Fig 2 shows the sensitivity of $I^0_H/N_H$ and $I_H(2020)/I^0_H$ sorted in the order of parameters' influence. Although the analysis is done for all parameters, the ones with negligible sensitivity values < 0.01 are left out of the plots.

We first look at the baseline model without asymptomatic population by setting $(p_{bs}, \omega^b_H) = (1, 0)$, so that there are no skin-only infections and no self-cure. Our results show the endemic equilibrium condition is almost invariant to many parameters and the dominating parameters are the basic reproduction number, $R_0$, relative bites taken on high-risk humans, $r$, and proportion of low-risk people, $k_1$. As we would expect, more parameters, such as the ones characterising passive detection rates and their change over time, contribute to the disease dynamics, here measured by the relative change in prevalence over time $I_H(2020)/I^0_H$. Another interesting remark is that our real-world observables (e.g. case reporting) are, in general, mainly sensitive to the parameter we had characterised as "fitted parameters" of the model and rather vary little with changes to the fixed parameters.

Performing the sensitivity analysis for the asymptomatic model shows the endemic equilibrium is not influenced much by asymptomatic parameters, however, these parameters play considerable roles in the system dynamics. As shown in the example of $I_H(2020)/I^0_H$ in Fig 2, the proportion of infections progressing directly to blood infections rather than skin-only infections, $p_{bs}$, is found to be the most decisive parameter, even compared to $R_0$.

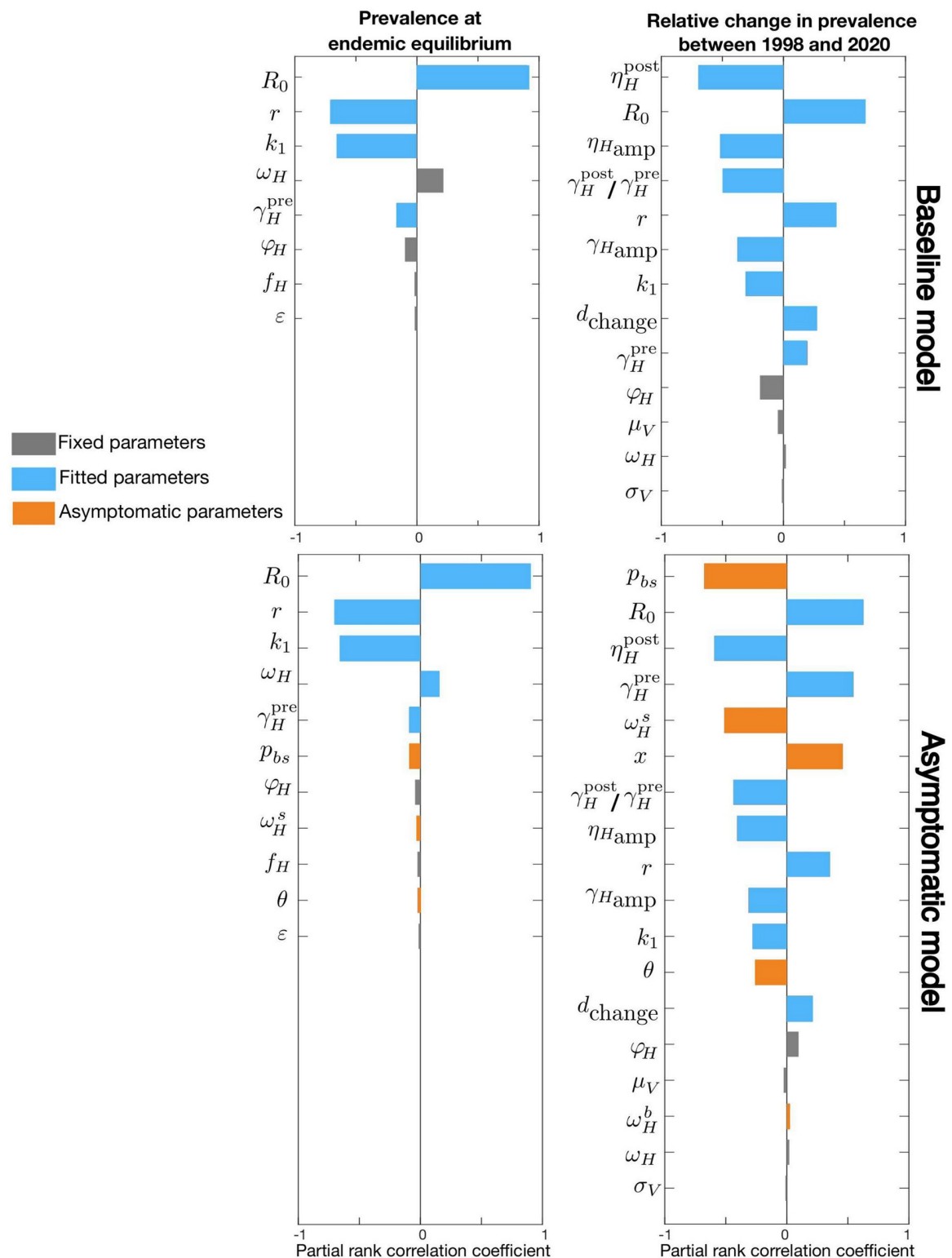

**Fig 2. Sensitivity analysis in terms of model parameters.** Sensitivity of total human prevalence, $I_H/N_H$, analysed for the whole parameter space of the baseline model (first row) and the asymptomatic model (second row). The first column corresponds to the endemic equilibrium configuration, $I_H^0/N_H$, and second column represents the relative change in prevalence between 1998 and 2020, $I_H(2020)/I_H^0$. We do not show parameters with sensitivity values below 1% and sort the parameters by their absolute sensitivity value in each plot. Fixed, fitted, and asymptomatic parameters are shown in gray, blue, and orange respectively.

## Endemic equilibrium of the asymptomatic model

After performing the general sensitivity analysis for our model to all parameters, we carefully study how endemic equilibrium is influenced by individual parameters corresponding to the asymptomatic infection. We therefore choose a set of parameters, here the one with the highest likelihood achieved by fitting the model to the available data in a moderate-risk setting in the DRC—a health zone named Mosango in Kwilu province. We then systematically analyse how each parameter of the asymptomatic model influences the prevalence at the endemic equilibrium.

To focus on this effect, we keep all the fitted and fixed parameters the same as as the baseline model and change the asymptomatic parameters. In our model the basic reproduction number, $R_0$, is considered as a fitted parameter because of a prior knowledge coming from data-based estimates. In that representation the effective tsetse density, $m_{eff}$, is calculated after fitting using the posterior estimates of $R_0$ (more details in the SI). The prevalence is then determined at the endemic equilibrium given $R_0$ and $m_{eff}$. In fact, the basic reproduction number, the prevalence, and the effective tsetse density are conjugate and fixing one can impose the other two. We therefore consider three different cases where one of these quantities, in addition to other parameters of the baseline model, is fixed and analyse each case separately.

In the first case, the basic reproduction number, $R_0$, is fixed. As shown in Fig 3, endemic prevalence increases as the proportion of skin infection, $p_{bs}$, decreases. The prevalence is also influenced by other asymptomatic transition rate parameters, $\omega_H^s$ and $\theta$, but it is not highly influenced by the relative infectivity of skin-only infections, $x$. Apart from the total prevalence $I_H/N_H$, we also look at $u\gamma_H(Y)I_{2H}$ as a measure of observable incidence through passive case detection since our model assumes only second stage infected humans can be diagnosed at the endemic equilibrium condition. Our stage two passive case detection incidence would be $u\gamma_H(Y)I_{2H}$ per unit time, and hence varies proportional to $I_{2H}$ for fixed reporting rate, $u$, and $\gamma_H(Y = 1997)$. In contrast to the total prevalence, the observable incidence is expected to be smaller in the asymptomatic model compared the baseline model. This effect is enhanced by decreasing $p_{bs}$, $\theta$, and $\omega_H^s$, however, the observed incidence remains unchanged as a function of $x$. This figure also shows how $m_{eff}$ needs to be adjusted to keep $R_0$ constant.

In the second case we fix observable incidence, $u\gamma_H(Y)I_{2H}$, as the baseline model and measure the corresponding $R_0$ and $m_{eff}$ for each choice of asymptomatic parameter set (Fig 4). The total prevalence in the asymptomatic model changes in a qualitatively similar manner to case 1, however within a broader range. $R_0$ follows the total prevalence and $m_{eff}$ remains very similar to the first case as well.

The third case describes the condition when the effective tsetse density, $m_{eff}$, is kept constant as the baseline model and it displays qualitatively different results from cases 1 and 2. In this case, as shown in Fig 5, when the relative infectiousness of skin-only infections, $x$, is below a threshold, the basic reproduction number is smaller than 1 and the endemic steady state vanishes. This threshold is dependent on the other model parameters, as per the basic reproduction number calculation in the SI. Above the threshold, the reproduction number and prevalence increase with $x$. The $x$ threshold is given by the parameters and mainly controlled by $\omega_H^s$.

## Infection dynamics in the asymptomatic model

Our sensitivity analysis suggested asymptomatic infection can play a significant role in the dynamics of the infection. In this section, we study that effect more carefully by solving equations of the gHAT dynamics as described in Eqs 1 and 2 over time.

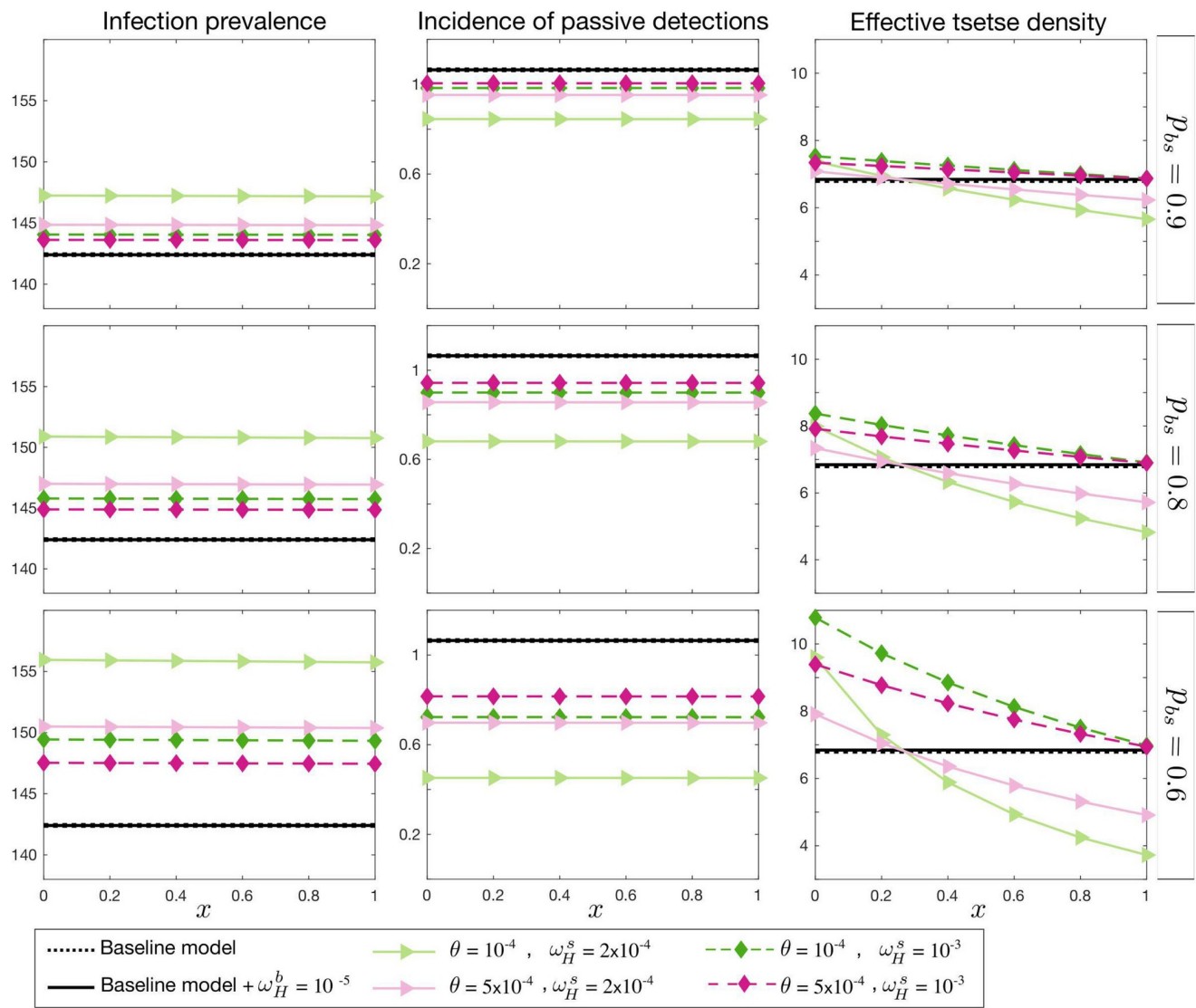

**Fig 3. Endemic equilibrium with fixed $R_0$.** Total prevalence, $I_H$, observable incidence, $u\gamma_H(Y)I_{2H}$, and the effective tsetse density, $m_{eff}$, are plotted as a function of $x$, the relative infectiousness of skin-only infections compared to blood infections. The proportion of blood to skin-only infections, $p_{bs}$, is fixed for each row and different values of the self-curing skin-only infection rate, $\omega_H^s$, and the skin-only to blood infection progression rate, $\theta$, are shown with various colours. There is no skin-only infection, $p_{bs} = 1$, for the black lines; the dashed line represents the baseline model (with no asymptomatic components) and the solid line represents the baseline model but with the possibility of self cure from first-stage blood infection, $I_{1H}^b$, at a rate $\omega_H^b = 10^{-5}$. The prevalence and incidence are given per 10,000 people.

Similar to the endemic equilibrium analysis, we start from the baseline model using the parameters with the most likelihood of all posterior parameter sets found by fitting the model to data for the health zone named Mosango [30] and setting the probability of progressing to blood infection, $p_{sb}$, to 1 and the self-cure from blood infection, $\omega_H^b$, to 0. We compare the results of the baseline model with the asymptomatic model for three cases introduced in the previous section (fixing either $R_0$, the observable passive incidence, or $m_{eff}$) by looking at the total prevalence ($I_H/N_H$) and reported cases by year for each of the different case detection modalities (active screening of the low-risk population or passive case detection dependent on self-presentation following symptoms). Fig 6 shows the results for a 20% probability of skin-

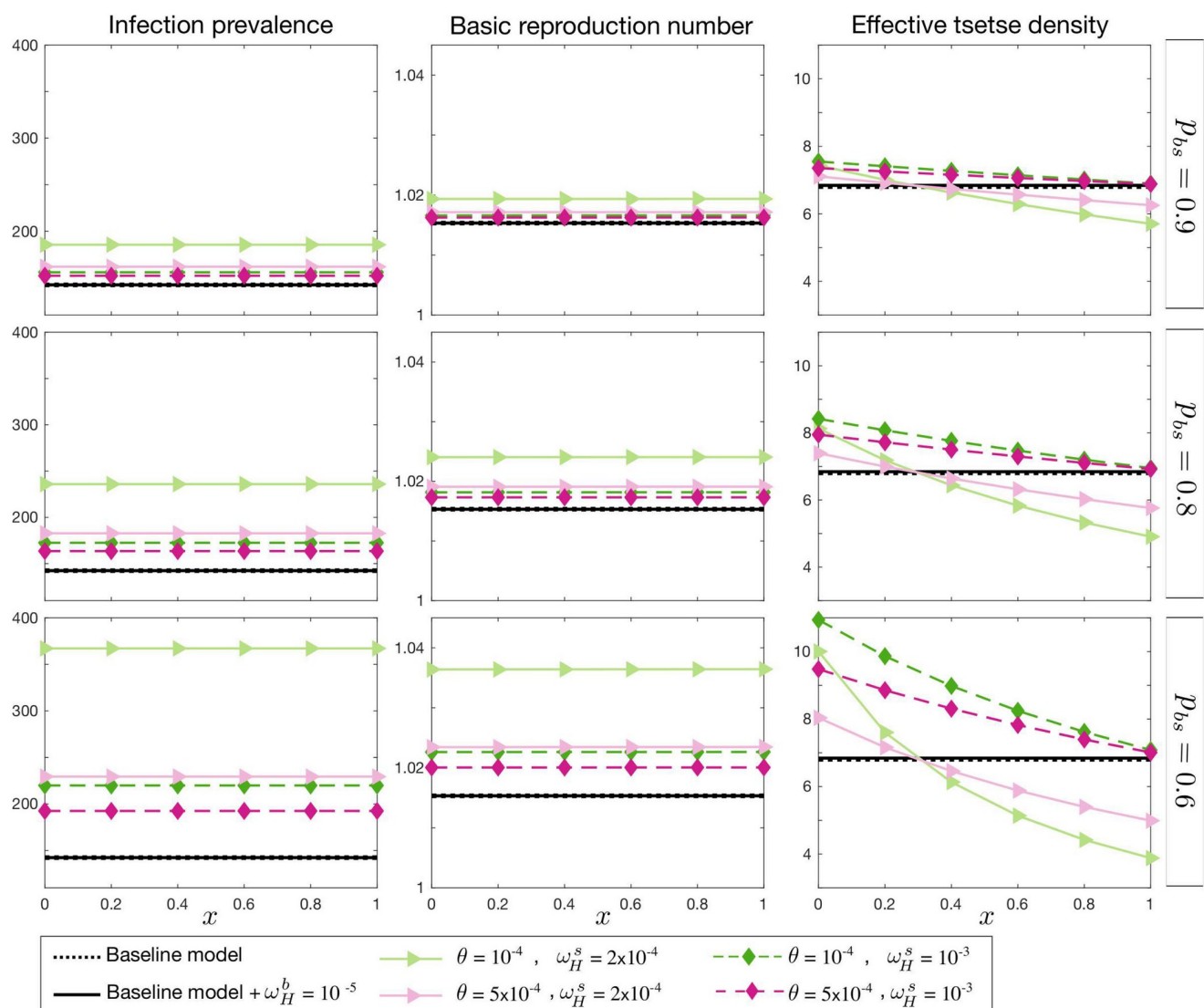

**Fig 4. Endemic equilibrium with observable incidence $u\gamma_H(Y)I_{2H}$ fixed.** Total prevalence, $I_H$, the basic reproduction number $R_0$, and the effective tsetse density, $m_{eff}$, are plotted as a function of $x$, the relative infectiousness of skin-only infections compared to blood infections. The proportion of blood to skin-only infections, $p_{bs}$, is fixed for each row and different values of the self-curing skin-only infection rate, $\omega_H^s$, and the skin-only to blood infection progression rate, $\theta$, are shown with various colours. There is no skin-only infection, $p_{bs} = 1$, for the black lines; the dashed line represents the baseline model (with no asymptomatic components) and the solid line represents the baseline model but with the possibility of self cure from first-stage blood infection, $I_{1H}^b$, at a rate $\omega_H^b = 10^{-5}$. The prevalence and incidence are given per 10,000 people.

only infection and three different choices of relative transmission probability from skin population, $x$, when the progression rates from early stage blood or skin-only infection or between them, $\omega_H^b = 10^{-5}, \omega_H^s = 2 \times 10^{-4}, \theta = 10^{-4}$, are fixed. Figures with other values of probability of skin-only infection are included in "S1" and "S2" Figs.

In case 1 and 2, when either of the basic reproduction number $R_0$ or the observable incidence is fixed at the initial endemic equilibrium, we see the asymptomatic infection changes the slope of prevalence decline significantly even when skin-only infected people have comparatively low infectiousness ($x = 0.2$). Both active and passive reported cases remain considerable in the asymptomatic model for years after it reaches threshold of 1 per 10,000 in the baseline model for these cases.

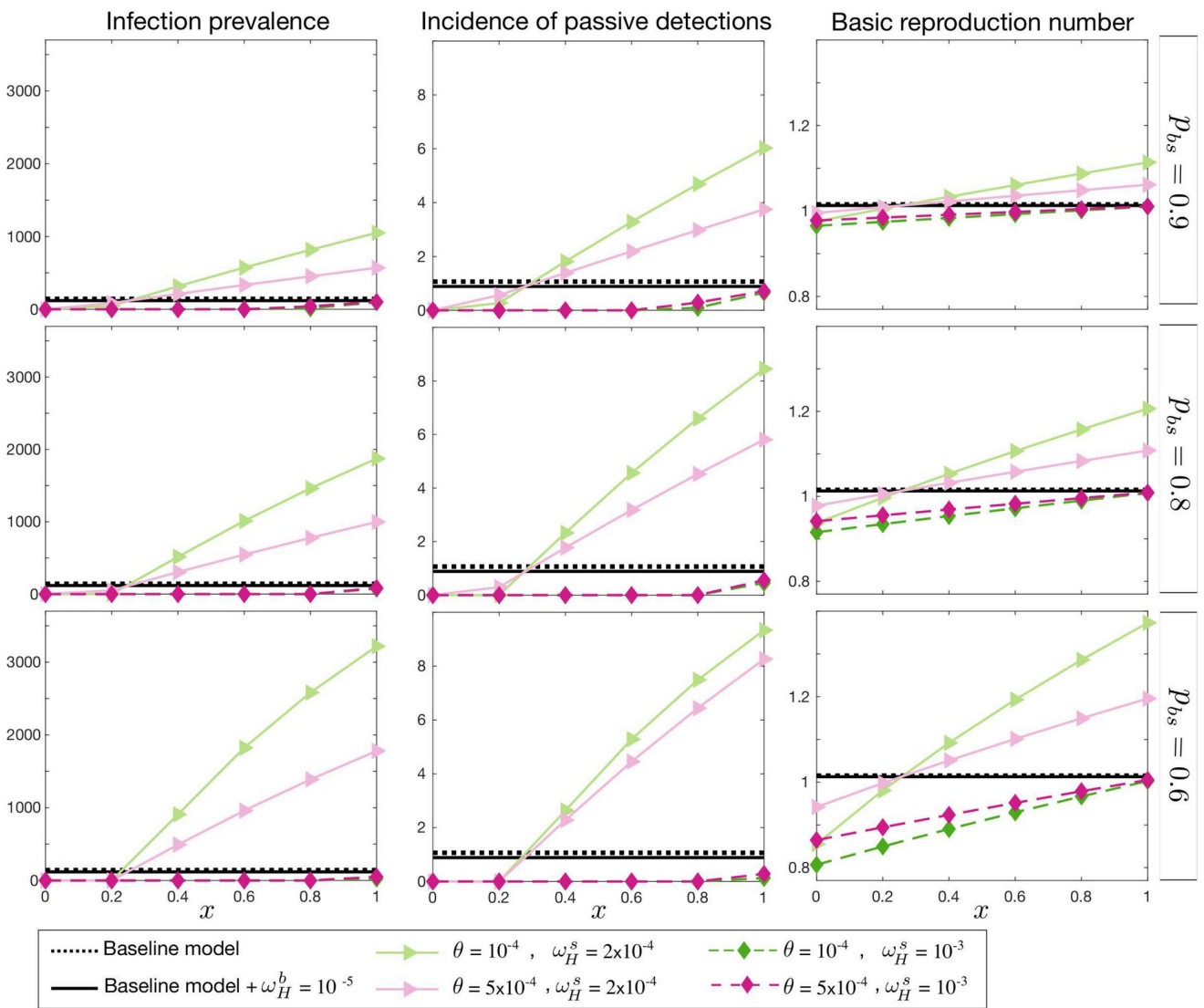

**Fig 5. Endemic equilibrium with fixed effective tsetse density, $m_{eff}$.** Total prevalence, $I_H$, observable incidence, $u\gamma_H(Y)I_{2H}$, and the basic reproduction number, $R_0$, are plotted as a function of $x$, the relative infectiousness of skin-only infections compared to blood infections. The proportion of blood to skin-only infections, $p_{bs}$, is fixed for each row and different values of the self-curing skin-only infection rate, $\omega_H^s$, and the skin-only to blood infection progression rate, $\theta$, are shown with various colours. There is no skin-only infection, $p_{bs} = 1$, for the black lines; the dashed line represents the baseline model (with no asymptomatic components) and the solid line represents the baseline model but with the possibility of self cure from first-stage blood infection, $I_{1H}^b$, at a rate $\omega_H^b = 10^{-5}$. The prevalence and incidence are given per 10,000 people.

Case 3, with $m_{eff}$ fixed according to the baseline model predicts very different dynamics. In particular for larger values of skin-only infectiousness, $x$, the basic reproduction number is relatively high ($\sim 1.3$) at the endemic equilibrium configuration, and it may reach a new, non-zero equilibrium in the long-term with a lower transmission level compared to the initial endemic equilibrium. However, for $x = 0.2$ transmission vanishes since the basic reproduction number is smaller than 1, in the same manner that was demonstrated by our endemic equilibrium results in the previous section. It is noted that, we may see a small bump in passive cases reported around the year $d_{change}$ in all cases which is a consequence of enhancing the passive detection rates $\eta_H(Y)$ and $\gamma_H(Y)$ over time.

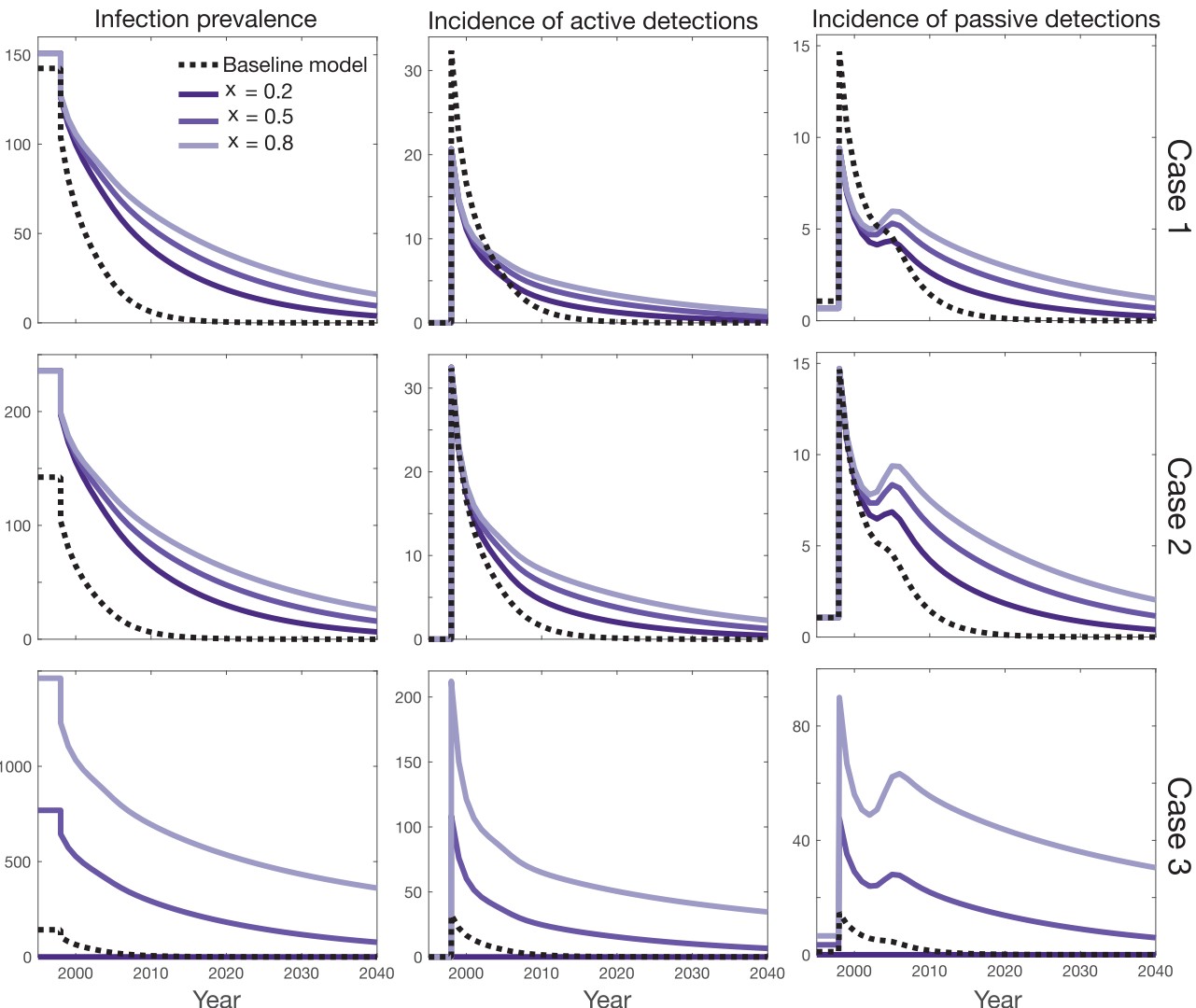

**Fig 6. Influence of asymptomatic infection on infection dynamics.** Total prevalence, active cases reported (without false-positive), and passive cases reported are plotted over the years 1998 to 2040. Each row corresponds to fixing either: (case 1) the basic reproduction number, $R_0$, (case 2) the observable incidence of passive detections, or (case 3) the effective tsetse density, $m_{eff}$. Different purple shades represent different values of relative skin-only infection infectiousness, $x$, when $p_{bs} = 0.8$, $\theta = 10^{-4}$, $\omega_H^b = 10^{-5}$, and $\omega_H^s = 2 \times 10^{-4}$ are fixed. The baseline model is shown as a dashed black line in each subplot ($p_{bs} = 0$ and $\omega_H^b = 0$). The prevalence and incidence are given per 10,000 people.

## Discussion

In this manuscript we presented a mathematical framework to study the potential impact of different types of asymptomatic infection in the transmission and dynamics of the sleeping sickness. We presented a model that takes into account asymptomatic cases of skin-only and blood parasite infections. Although trypanosomes can be accumulated in other organs, but in this study we focused on skin infection since it is much better studied and it is likely that skin-only infections (with no detectable blood parasitaemia) could still result in transmission to tsetse [6, 7, 11]. Our model is therefore a minimal model formulation which could be extended to incorporate more sophisticated asymptomatic scenarios that bring additional complexity.

We performed sensitivity analysis for this model to study how different parameters influence the model outputs, including in the baseline model with no skin-only infections and no

self-cure. This analysis provides insight into the comparative importance of each parameter and can thereby support which parameters to select for fitting to data. The parameter ranges included here could also help choose appropriate priors for parameters selected for fitting in subsequent analyses, as the ranges were guided by a limited but informative selection of previous studies from the laboratory and field which reported on the natural history of infection and the relative infectivity of gHAT in skin-only and blood infections.

The sensitivity results suggest the asymptomatic parameters can play significant roles in model dynamics; in particular the proportion of infections which are initially blood rather than skin-only infections ($p_{bs}$), the self-cure rate from skin-only infection ($\omega_H^s$) and the relative infectiousness of skin-only infections ($x$) were highlighted as having substantive impact on infection and reporting trends over time. We therefore systematically studied how each of these parameters influences both the endemic equilibrium and the transmission dynamics under medical control interventions (active and passive screening). Unsurprisingly the decay to zero can be much slower under the model with asymptomatic infection compared to our baseline model, due to the increased infectious pressure from the skin-only human reservoir. For some parameter set choices, it was even possible that following rapid decline, the dynamical system reached a new, lower endemic equilibrium, which would be particularly worrying in the context of targeting gHAT for elimination of transmission by 2030.

The results presented here are not intended to provide policy guidance, as they are not fitted to case data, but provide qualitative descriptions of the impact of choosing such a model and a quantitative ranking of the importance of the model parameters to impact infection dynamics. Future studies will use the model presented here to fit the new asymptomatic parameters at the same time as the previously identified list of fitted parameters, across a range of settings to establish the impact the potential asymptomatic infections have on long-term elimination prospects under different intervention strategies. It will require careful consideration of available evidence to provide meaningful priors for a better estimate of the new parameters.

It should be emphasised that our model is based on the current protocols of medical diagnoses and treatment. For instance, the failure of active screening in identifying skin-only infections due to the negative parasitological tests done on blood or plasma is a decisive assumption for the model and was chosen to be in line with current WHO recommendation [35]. New studies offer the hope that alternative medicines—such as acoziborole, which is in phase 3 clinical trials [36]—could significantly change the paradigm of gHAT active screening if it were approved for all seropositive individuals, or even all at-risk individuals. In such as scenario, skin-only infections, which could have played some role in capping the potential impact of screening programmes, could then be eligible for treatment and this could rapidly curtail infections resulting from this reservoir. This model framework would be well-suited to examine the potential impact of new treatment options.

The new framework presented is deterministic in nature and therefore, if in order to provide predictions for elimination years, a proxy threshold must be used to specific when zero infection is expected. Our previous analysis suggested it is possible to approximate stochastic, local elimination using a proxy threshold, however a constant value threshold cannot be used for different choices of model parameters and careful analysis would be required for further quantitative estimates [37]. Alternatively a stochastic version of this model could be used in a tau-leaping or Gillespie approach to simulate end-game dynamics, as has been used for the baseline model [38–41].

Similar to the challenge of asymptomatic human infection, there have been debates on the impact of animal reservoirs to sustain transmission of gHAT. On-going work by our group is separately assessing evidence for animal transmission across different geographic locations. The presented asymptomatic model can be modified as well to take into account such animal reservoir.

This study represents an important first step in development and analysis of a model which can describe different types of asymptomatic human infection. It can be used to explore the potential dynamics of gHAT infection under different types of screening and treatment of cases. It is now well placed to be used in future studies to determine location-specific impact of asymptomatic infections on long-term dynamics and the possible routes to elimination. In particular predictions can be made under continuation of current strategy or the use of supplemental strategies, which have the ability to better target skin-only infection or transmission such as acoziborole or vector control.

## Supporting information

**S1 Methods. This file provides an additional description of our methods to estimate parameters, compare models and compute the endemic equilibrium.**
(PDF)

**S1 Fig. Influence of asymptomatic infection on infection dynamics.** Total prevalence, active cases reported (without false-positive), and passive cases reported are plotted over the years 1998 to 2040. Each row corresponds to fixing either: (case 1) the basic reproduction number, $R_0$, (case 2) the observable incidence of passive detections, or (case 3) the effective tsetse density, $m_{eff}$. Different purple shades represent different values of relative skin-only infection infectiousness, $x$, when $p_{bs} = 0.9$, $\theta = 10^{-4}$, $\omega_H^b = 10^{-5}$, and $\omega_H^s = 2 \times 10^{-4}$ are fixed. The baseline model is shown as a dashed black line in each subplot ($p_{bs} = 0$ and $\omega_H^b = 0$). The prevalence and incidence are given per 10,000 people.
(EPS)

**S2 Fig. Influence of asymptomatic infection on infection dynamics.** Total prevalence, active cases reported (without false-positive), and passive cases reported are plotted over the years 1998 to 2040. Each row corresponds to fixing either: (case 1) the basic reproduction number, $R_0$, (case 2) the observable incidence of passive detections, or (case 3) the effective tsetse density, $m_{eff}$. Different purple shades represent different values of relative skin-only infection infectiousness, $x$, when $p_{bs} = 0.7$, $\theta = 10^{-4}$, $\omega_H^b = 10^{-5}$, and $\omega_H^s = 2 \times 10^{-4}$ are fixed. The baseline model is shown as a dashed black line in each subplot ($p_{bs} = 0$ and $\omega_H^b = 0$). The prevalence and incidence are given per 10,000 people.
(EPS)

**S1 Analysis Code. Our code to analyse the dynamics of our asymptomatic model for any parameter set.**
(CPP)

## Acknowledgments

Whilst no human case data were specifically used in this study, the results presented relied on previous model fitting [30], which utilised data collected from the Democratic Republic of Congo (DRC). The authors thank Programme National de Lutte contre la Trypanosomiase Humaine Africaine (PNLTHA) of DRC and its director Dr Erick Mwamba Miaka for original data collection and WHO for data access (in the framework of the WHO HAT Atlas [18]).

## Author Contributions

**Conceptualization:** Maryam Aliee, Matt J. Keeling, Kat S. Rock.

**Formal analysis:** Maryam Aliee.

**Funding acquisition:** Matt J. Keeling, Kat S. Rock.

**Investigation:** Maryam Aliee.

**Methodology:** Maryam Aliee, Matt J. Keeling, Kat S. Rock.

**Software:** Maryam Aliee, Kat S. Rock.

**Supervision:** Matt J. Keeling, Kat S. Rock.

**Visualization:** Maryam Aliee.

**Writing – original draft:** Maryam Aliee, Kat S. Rock.

**Writing – review & editing:** Maryam Aliee, Matt J. Keeling, Kat S. Rock.

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
