## [Decision Letter · Decision Letter 0]

9 Jul 2021

Dear Dr Aliee,

Thank you very much for submitting your manuscript "Modelling to explore the potential impact of asymptomatic human infections on transmission and dynamics of African sleeping sickness" for consideration at PLOS Computational Biology. As with all papers reviewed by the journal, your manuscript was reviewed by members of the editorial board and by several independent reviewers. The reviewers appreciated the attention to an important topic. Based on the reviews, we are likely to accept this manuscript for publication, providing that you modify the manuscript according to the review recommendations.

Both reviewers raise legitimate issues that require the attention of the authors. In particular, we would like to see an expanded discussion on the final choice of model structure and the motivations that inhibited the authors from also addressing the role of animal reservoirs.

Sincerely,

Claudio José Struchiner, M.D., Sc.D.

Associate Editor

PLOS Computational Biology

Virginia Pitzer

Deputy Editor-in-Chief

PLOS Computational Biology

[LINK]

Both reviewers raise legitimate issues that require the attention of the authors. In particular, we would like to see an expanded discussion on the final choice of model structure and the motivations that inhibited the authors from also addressing the role of animal reservoirs.

Reviewer's Responses to Questions

**Comments to the Authors:**

Reviewer #1: The manuscript is a well written and timely description of a model for HAT that can be used to examine the role that latent carriers may play in the maintenance of disease foci. The model is adapted from recent work and the additional parameters included are generally clearly described and justified. Sensitivity testing and infection simulations suggest that asymptomatic carriers of HAT can impact infection dynamics over time. The model serves as a useful starting point to estimate the impact of asymptomatic HAT and the treatment of such cases as the disease approaches elimination.

I only have a few minor points:

- I think more detail is required for describing how the ranges for the self-cure rate of skin-only infections (ωsH) and skin-only to blood infection progression rate (θ) were derived. As mentioned in the manuscript, parasites can sequester in compartments other than the skin. Such individuals would still be serologically positive and defined as asymptomatic but would not contribute to transmission. It would be useful to clearly state how the initial ranges for these parameters were established and how the distinction between asymptomatic skin infections and asymptomatic infections in other organs (or indeed CATT false positives) can be accounted for.

- Similarly, while skin-only parasite infections being the primary manifestation of asymptomatic cases is a compelling argument (as stated at line 68-70), I do not believe that the evidence is strong enough to preclude other compartments serving as an anatomical reservoir in asymptomatic infections, particularly pre-latent cases that go on to develop symptoms. This statement should probably be softened or rephrased slightly.

-While I appreciate that the manuscript is explicitly only examining human cases and animal reservoirs are not considered, this is only useful in a few HAT foci, if any. The authors state that animals could be easily added but I wonder if showing both the current model with no/few animal reservoirs (similar to the situation in Guinea) and another with well described animal reservoirs (such as those found in Cameroon) would be useful for the non-modelling community. Do the authors have a compelling argument for not including the animal reservoir? I note the recent publication of models to examine the role that animal reservoirs may play in T. b. gambiense HAT, and now this examining latent human infections, but none that consider both.

Reviewer #2: This manuscript entitled « Modelling to explore the potential impact of asymptomatic human infections on transmission and dynamics of African sleeping sickness” by Maryam Aliee et al. is presenting a new g-HAT model of transmission which, for the first time includes, human asymptomatic infections parameters. Whereas the existence of asymptomatic infections in HAT is now widely recognized, the exact role of these individuals on transmission is largely unknown but is receiving increased attention with the recent discovery of skin-dwelling trypanosomes both in mice experimental models and in HAT patients and serological suspects otherwise testing negative in blood. The availability of such models integrating asymptomatic or skin-only infected individuals is thus timely and will likely be key in addressing not only biological questions such as the infectiousness of such individuals (that is very difficult to assess in the field) but also, as very well illustrated in the paper, the potential impact such individuals could have on the infection dynamic and the 2030 “0” transmission objective. To my point of view the paper is nicely written and the scientific approach taken to build-up this new model (including sensitivity analysis, influence of the new parameters on the endemic equilibrium and on the infection dynamics) is clearly explained and understandable even to none specialists. With my knowledge of g-HAT epidemiology, I also agree with the new “asymptomatic parameters” that were added to the base line model (fitted to the Mosango health zone data in DRC) that enable I think to capture well the information from asymptomatic carriers. It is to my point of view too early to withdraw firm conclusions on the impact asymptomatic carriers may have in terms of progression to disease elimination or “0” transmission as further work is required to better assess in endemic areas the proportion of skin-only carriers and the length of carriage (all parameters that may greatly vary in between foci and that were not available for the Mosango health zone). It is however well stated in the discussion that the next step will be to fit the new asymptomatic parameters to field collected data in a number of settings. Forthcoming data from the prevalence of skin carriage in different endemic areas should enable to improve parameter estimation locally and model predictions. As to my point of view this manuscript represents an important step in the mathematical modelling of g-HAT transmission, I recommend it for publication in PLOS Computational Biology. I nevertheless have some comments below to which authors should respond and/or modify their manuscript.

Major comment

In the model the population is divided into low-risk and high-risk humans. Throughout the manuscript it is said that “we assume that high risk individuals are unlikely to participate in active screening, but low risk individuals may present randomly and according to the coverage of active screening in each year” (line 91, line 115, Fig 1 legend, line 311). Authors need to clarify this point as it is usually the reverse that is done. Active screening campaigns are carried out in high risk populations (villages where most HAT cases are coming from and where transmission is likely active) whereas HAT cases are only detected passively, based on symptoms, in the low risk population. It is thus not clear to me if this is a mistake in the text, if the model was built this way, or if this was to take into account the fact that indeed a fraction of the population at high risk (like young active males) participate less in active screening.

Minor comments

Material and Methods -Estimation of asymptomatic parameters: These estimations are of course very difficult to make due to the scarcity of data. I was surprized that concerning the estimation of the proportion of blood/skin only infections (Pbs) the authors only considered the study by Capewell et al. reporting on 6/1121 trypanosome positive archived skin biopsies collected in DRC for onchocerciasis diagnosis. This study was not designed at all to estimate the prevalence of skin carriage but rather to prove it existed. Another recent study led in Guinea is describing the prevalence of skin carriage (IHC and PCR detection) and provide follow-up data from these subjects. Although this paper by Camara et al., is already cited in the manuscript, I would advise the author to discuss also, in this part of the material and method section or elsewhere, how the Guinean data fit into the range they tested for Pbs, and other parameters such the self-cure of skin-only (WsH) or the transition rate from skin-only to blood infections (θ).

Figure 1: legend says: «This marked on the diagram as a grey box”, but I couldn’t see any grey box

Figure 4 and 5 are not referred in the text.

**Have the authors made all data and (if applicable) computational code underlying the findings in their manuscript fully available?**

Reviewer #1: Yes

Reviewer #2: Yes

PLOS authors have the option to publish the peer review history of their article (what does this mean?). If published, this will include your full peer review and any attached files.

Reviewer #1: No

Reviewer #2: **Yes: **Bruno Bucheton

Figure Files:

Data Requirements:

Reproducibility:

References:

---

## [Decision Letter · Decision Letter 1]

20 Aug 2021

Dear Dr Aliee,

We are pleased to inform you that your manuscript 'Modelling to explore the potential impact of asymptomatic human infections on transmission and dynamics of African sleeping sickness' has been provisionally accepted for publication in PLOS Computational Biology.

Best regards,

Claudio José Struchiner, M.D., Sc.D.

Associate Editor

PLOS Computational Biology

Virginia Pitzer

Deputy Editor-in-Chief

PLOS Computational Biology

Reviewer's Responses to Questions

**Comments to the Authors:**

Reviewer #1: The author's have addressed my minor comments and provided additional information to explain the initial parameter ranges used.

**Have the authors made all data and (if applicable) computational code underlying the findings in their manuscript fully available?**

Reviewer #1: Yes

PLOS authors have the option to publish the peer review history of their article (what does this mean?). If published, this will include your full peer review and any attached files.

Reviewer #1: No

---

## [Editor Report · Acceptance letter]

3 Sep 2021

PCOMPBIOL-D-21-00839R1 

Modelling to explore the potential impact of asymptomatic human infections on transmission and dynamics of African sleeping sickness

Dear Dr Aliee,

I am pleased to inform you that your manuscript has been formally accepted for publication in PLOS Computational Biology. Your manuscript is now with our production department and you will be notified of the publication date in due course.

With kind regards,

Amy Kiss
